# Diagnostic and Therapeutic Algorithm for Appendiceal Tumors and Pseudomyxoma Peritonei: A Consensus of the Peritoneal Malignancies Oncoteam of the Italian Society of Surgical Oncology (SICO)

**DOI:** 10.3390/cancers15030728

**Published:** 2023-01-24

**Authors:** Marco Vaira, Manuela Robella, Marcello Guaglio, Paola Berchialla, Antonio Sommariva, Mario Valle, Enrico Maria Pasqual, Franco Roviello, Massimo Framarini, Giammaria Fiorentini, Paolo Sammartino, Alba Ilari Civit, Andrea Di Giorgio, Luca Ansaloni, Marcello Deraco

**Affiliations:** 1Unit of Surgical Oncology, Candiolo Cancer Institute, FPO—IRCCS, 10060 Candiolo, Italy; 2Peritoneal Surface Malignancies Unit, Fondazione Istituto Nazionale Tumori IRCCS Milano, 20133 Milano, Italy; 3Department of Clinical and Biological Sciences, Centre for Biostatistics, Epidemiology and Public Health (C-BEPH), University of Torino, 10124 Torino, Italy; 4Advanced Surgical Oncology Unit, Surgical Oncology of the Esophagus and Digestive Tract, Veneto Institute of Oncology IOV-IRCCS, t, 35100 Padova, Italy; 5Peritoneal Tumours Unit, IRCCS Regina Elena National Cancer Institute, 00144 Rome, Italy; 6AOUD Center Advanced Surgical Oncology, DAME University of Udine, 33100 Udine, Italy; 7Unit of General Surgery and Surgical Oncology, Department of Medicine, Surgery, and Neurosciences, University of Siena, 53100 Siena, Italy; 8Surgery and Advanced Oncological Therapy Unit, Ospedale “GB.Morgagni-L.Pierantoni”—AUSL Forlì, 47122 Forlì, Italy; 9Italian Network of International Clinical Hyperthermia Society Coordinator, 48121 Ravenna, Italy; 10CRS and HIPEC Unit, Pietro Valdoni, Umberto I Policlinico di Roma, 00161 Roma, Italy; 11Surgical Unit of Peritoneum and Retroperitoneum, Fondazione Policlinico Universitario A. Gemelli—IRCCS, 00168 Rome, Italy; 12Unit of General Surgery, San Matteo Hospital, 27100 Pavia, Italy

**Keywords:** PMP, pseudomyxoma peritonei, peritoneal surface malignancies, HIPEC, cytoreductive surgery, DPAM, PMCA, LAMN, HAMN, peritoneal carcinomatosis

## Abstract

**Simple Summary:**

Pseudomyxoma peritonei (PMP) is an uncommon pathology, and its rarity causes a lack of scientific evidence, precluding the design of a prospective trial. A diagnostic and therapeutic algorithm (DTA) is necessary in order to standardize disease treatment while balancing optimal patient management and the correct use of resources. The Consensus of the Italian Society of Surgical Oncology (SICO) Oncoteam aims at defining a diagnostic and therapeutic pathway for PMP and appendiceal primary tumors applicable in Italian healthcare.

**Abstract:**

**Aim:** Pseudomyxoma peritonei (PMP) is an uncommon pathology, and its rarity causes a lack of scientific evidence, precluding the design of a prospective trial. A diagnostic and therapeutic algorithm (DTA) is necessary in order to standardize the disease treatment while balancing optimal patient management and the correct use of resources. The Consensus of the Italian Society of Surgical Oncology (SICO) Oncoteam aims at defining a diagnostic and therapeutic pathway for PMP and appendiceal primary tumors applicable in Italian healthcare. **Method:** The consensus panel included 10 delegated representatives of oncological referral centers for Peritoneal Surface Malignancies (PSM) affiliated to the SICO PSM Oncoteam. A list of statements regarding the DTA of patients with PMP was prepared according to recommendations based on the review of the literature and expert opinion. **Results:** A consensus was obtained on 33 of the 34 statements linked to the DTA; two flowcharts regarding the management of primary appendiceal cancer and peritoneal disease were approved. **Conclusion:** Currently, consensus has been reached on pathological classification, preoperative evaluation, cytoreductive surgery technical detail, and systemic treatment; some controversies still exist regarding the exclusion criteria for HIPEC treatment. A shared Italian model of DTA is an essential tool to ensure the appropriateness and equity of treatment for these patients.

## 1. Introduction

Pseudomyxoma peritonei (PMP) is a rare peritoneal malignancy characterized by an effusion of mucinous or viscous ascites in the peritoneal cavity, associated or not with the presence of epithelial cells, whose degree of malignancy is variable. The incidence of PMP is estimated at 1–3 per million people annually [1]. It most commonly originates from the rupture of an appendiceal mucinous neoplasm; in rare cases, it can derive from a tumor of the ovary, cervix, or urachus.

The symptoms are heterogenous: often appendiceal cancers are found incidentally at radiological exam or laparoscopy/laparotomy for appendicitis. The most common symptoms are pain and abdominal distension; over time, this syndrome results in massive abdominal bloating and associated mechanical and functional gastrointestinal obstruction [2].

Although this peritoneal malignancy is minimally invasive and rarely causes hematogenous or lymphatic metastases, conventional surgical management of repeated interval debulking for symptomatic relief resulted in a low long-term survival expectation [3,4]. In the early 1990s, the introduction of a combined approach based on the association of cytoreductive surgery (CRS), aiming for macroscopic complete tumor removal, and HIPEC (Hyperthermic Intraperitoneal Chemotherapy) has attracted increasing attention worldwide, has become the standard of treatment for PMP.

Given its rarity, randomized controlled trials on its management are lacking; in this scenario, diagnostic and therapeutic algorithms (DTA) are mandatory for several reasons. CRS and HIPEC are available in a limited number of specialized centers; moreover, most PMPs are diagnosed incidentally, often in an emergency setting. A guideline can help follow a correct diagnostic-therapeutic path and direct patients to referral centers.

## 2. Materials and Methods

The consensus panel included 10 delegated members of oncological referral centers affiliated with the Peritoneal Surface Malignancies Group of the Italian Society of Surgical Oncology (SICO). The recommendations are based on a review of the literature and expert opinion, as well as evidence synthesis. The statements covered the entire path from diagnosis through the decision-making process to treatment with curative and/or palliative intent for patients with appendiceal cancer and pseudomyxoma peritonei.

The topics of the statements were evaluated and modified during a first online meeting of the expert panel. The panel then divided the indications into two rounds using the RAND/UCLA Appropriateness Method (RAM) [5]. In the first round, the ratings were made individually and anonymously via the web, with no interaction among panelists. Appropriateness was evaluated on a scale of 1 to 9, where 1 means inappropriate and 9 means completely appropriate, and agreement was based on the interpercentile range (IPR) of 0.3–0.7. The appropriateness median score (AMS) for each statement was calculated based on how they were classified: appropriate (AMS in range 7–9), uncertain (AMS in range 4–6), and inappropriate (AMS in range 1–3).

In the second round, the panel members met on the web; each panelist received before the meeting a document showing the distribution of all the experts’ first round ratings. During the discussion, panelists debated the ratings, focusing on areas of disagreement; then the panel rerated each indication individually and anonymously. The same method was used for the two-round rating of the two flowcharts.

## 3. Results

The statements were related to different question points regarding the management of appendiceal tumors and pseudomyxoma peritonei, from diagnosis to treatment. All the statements were considered appropriate, with an AMS ranging between 7 and 9, and all the propositions were considered valid. Statement 26, although scored as appropriate (AMS 8), did not reach consensus with an IPR lower than the limit (6.7–8)—Table 1. The voting results are summarized in Table 2.

The two flowcharts about PMP management (Figure 1) and primary appendiceal cancer treatment (Figure 2) were considered well appropriate, with an AMS of 8 and 9, respectively, and an IPR of 8–9 for both.

## 4. Discussion

The DTA represents a systematic process, specific for disease and stage, that leads each specialist along the diagnostic-therapeutic path of the disease. The purpose of the DTAs is to increase the quality of assistance, equity in terms of access to care, and availability while improving outcomes and promoting patient safety through the use of the right resources.

The rarity of the pathology precluded the design of prospective trials. This consensus tried to define the best diagnostic and therapeutic algorithm for pseudomyxoma peritonei and appendiceal primary tumors. The vast majority of these statements are based on data obtained in clinical studies and experts’ recommendations on disease management.

There was widespread agreement that patients diagnosed with PMP should be evaluated at a peritoneal surface malignancy referral center. Most patients are referred from other surgical/gynecological, or emergency units. Referring teams should perform the minimal surgical procedures required to obtain a histopathological diagnosis or solve the emergency condition; extensive previous attempts to reduce tumor load were shown to have a negative impact on survival [6].

There has been considerable debate in the literature about the site of origin and pathological classification of PMP. These difficulties in the pathological classification of the clinical entity of PMP have led to ongoing confusion concerning the appropriate treatment.

In 1995, Carr et al. proposed a classification of appendicular mucinous tumors based on their review of 184 tumors into adenoma, mucinous tumors of uncertain malignant potential, and adenocarcinoma [7]. In the same year, Ronnett et al. classified patients into three groups based on the pathologic features of their peritoneal lesions: DPAM (Disseminated Peritoneal Adenomucinosis), PMCA (Peritoneal Mucinous Carcinomatosis), and PMCA-I/D (Peritoneal Mucinous Carcinomatosis Intermediate/Discordant) [8].

In 2003, Misdraji et al. subsequently proposed a two-tiered system using “low-grade appendiceal mucinous neoplasm” (LAMN) for all low-grade mucinous tumors of the appendix that do not demonstrate invasion of the appendiceal wall, either confined to the appendix or that have spread to the peritoneum; adenocarcinoma was reserved for tumors with either high-grade cytology or destructive invasion [9].

Other classifications of primary pathology and peritoneal diffusion have emerged over time [10,11].

At the PSOGI meeting in 2012 in Berlin, a generalized inhomogeneity was still present, and consequently, in 2016, a consensus was organized in order to standardize diagnostic terminology for appendiceal mucinous tumors with or without peritoneal disease; according to the panelists’ ratings, the PSOGI 2016 Consensus for histopathological classification should be adopted [12].

A preoperative work-up including serum markers (CEA, Ca19.9, and CA125), a CT scan, and a colonoscopy was rated as fully appropriate by the majority of the panelists (80%); in some studies, preoperative elevation of tumor markers was reported to be linked to the probability of complete cytoreduction [13] and strictly linked to outcome [14,15,16,17,18,19,20]. A CT scan should be the preferred diagnostic imaging technique [21,22]; MRI could be considered in selected cases [23,24,25].

Laparoscopy in order to obtain a histopathologic diagnosis and evaluate resectability may be considered [26,27,28]. Some authors suggested the use of a single port platform or the midline positioning of the trocars in order to allow, in the case of cytoreduction, the removal of the trocar site, avoiding potential seeding pathways. The panel’s recommendation is that the preoperative laparoscopic evaluation of patients with PMP should be done by a surgeon expert in PSM.

Appendectomy is curative in cases of incidental adenoma, hyperplastic polyp, or LAMN in the appendectomy specimen [29,30]. In cases of perforated LAMN with extra-appendicular cells, a CRS + HIPEC could be considered. Some authors suggested, on the basis of the low recurrence risk in patients with radically resected LAMN and limited peritoneal spread, clinical and radiological surveillance if localized cells or mucin are present outside the appendix [31].

In the case of a neuroendocrine tumor, a right-sided hemicolectomy should be considered if unfavorable prognostic factors are present [32]. The literature about HAMN and goblet cell carcinoma is scarce; panelists suggested a more aggressive approach similar to that of adenocarcinoma [33,34,35,36].

There was complete agreement in considering CRS + HIPEC as the first therapeutic option in patients with PMP [37,38,39]. Although gross small bowel involvement is a well-known predictor of unresectability [40,41,42], some concerns were raised about considering extensive small bowel involvement or mesenteric involvement with retraction an absolute contraindication for CRS + HIPEC; this is the panel’s only statement that was deemed invalid.

The main determinant of the outcome, beside histology, is the completeness of the cytoreduction [38]; however, in cases in which complete CRS may not be achievable, maximal tumor debulking might improve survival and quality of life [37,43,44,45].

The role of systemic treatment in the management of PMP is still poorly investigated: the small amount of available evidence shows no benefit from neoadjuvant chemotherapy neither in low-grade nor in high-grade PMP [46,47,48,49,50,51,52,53,54]; moreover, some studies report a worse survival rate in these patients [55]. Adjuvant systemic chemotherapy should not be considered in patients with low-grade PMP; in high-grade peritoneal disease, one large study showed a positive effect on overall survival, contrasting with the results of the other three monocentric studies [47,56,57,58].

Palliative systemic chemotherapy can be considered when surgery is not feasible for unresectable disease or poor general conditions [59,60]: despite the well-known unresponsiveness and chemoresistance of PMP cells to systemic treatment, a clinical response rate between 8 and 20% with a median overall survival of 25–26 months were reported [61,62,63,64,65]. A possible advantage can be given by anti-angiogenic treatment [27,44,66,67].

According to literature results, the mitomycin C (MMC) regimen is recommended in patients with PMP eligible for CRS + HIPEC [68,69], although there are still controversies regarding the concentration and the dosage [70,71]. Currently, different oxaliplatin-based regimens are used [72,73], despite the high rate of hemorrhagic complications [74,75]. A recent randomized controlled trial compared oxaliplatin (200 mg/m^2^) and MMC (40 mg) in closed HIPEC, evaluating toxicity, quality of life, and survival [76]: no significant difference between the two groups was reported. In conclusion, oxaliplatin can be used in HIPEC for patients with PMP instead of mitomycin C based on current clinical and pharmacological evidence.

Analogous recommendations were reported in a recently published consensus by members of the PSOGI Executive Committee on the management of appendiceal cancer and PMP; recommendations were provided based on three Delphi voting rounds with GRADE-based questions amongst a panel of 80 worldwide PMP experts [39].

## 5. Conclusions

The increasing network of PSM-specialized centers may help to consolidate the available data about this rare pathology, compensating for the lack of hard scientific evidence in PMP treatment. This DTA for the management of patients with appendiceal cancer and PMP is an important tool for healthcare providers to ensure the appropriateness and equity of treatment for these patients.

## Figures and Tables

**Figure 1 cancers-15-00728-f001:**
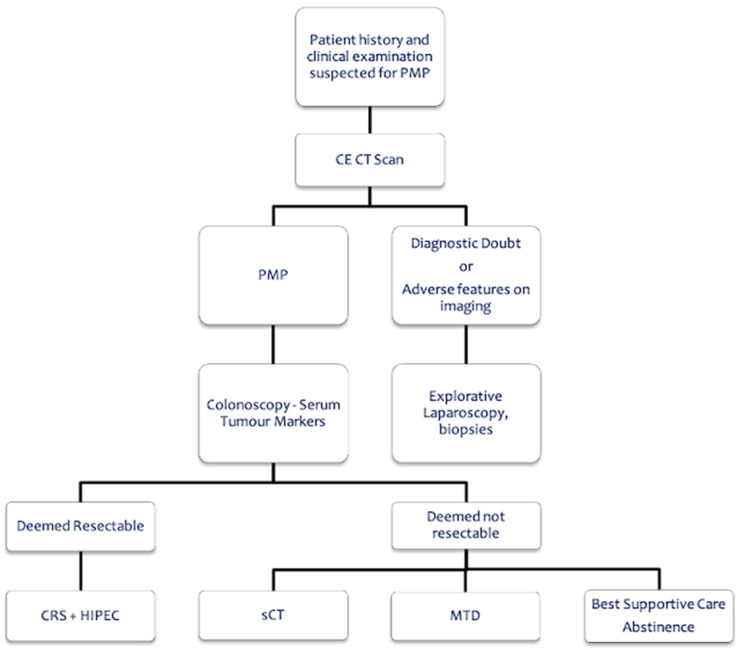
Pseudomyxoma peritonei management. CE CT Scan = Contrast Enhanced Computed Tomography; PMP = PseudoMyxoma Peritonei; CRS = Cytoreductive Surgery; HIPEC = Hyperthermic IntraPEritoneal Chemotherapy; sCT = Systemic ChemoTherapy; MTD = Maximal Tumor Debulking.

**Figure 2 cancers-15-00728-f002:**
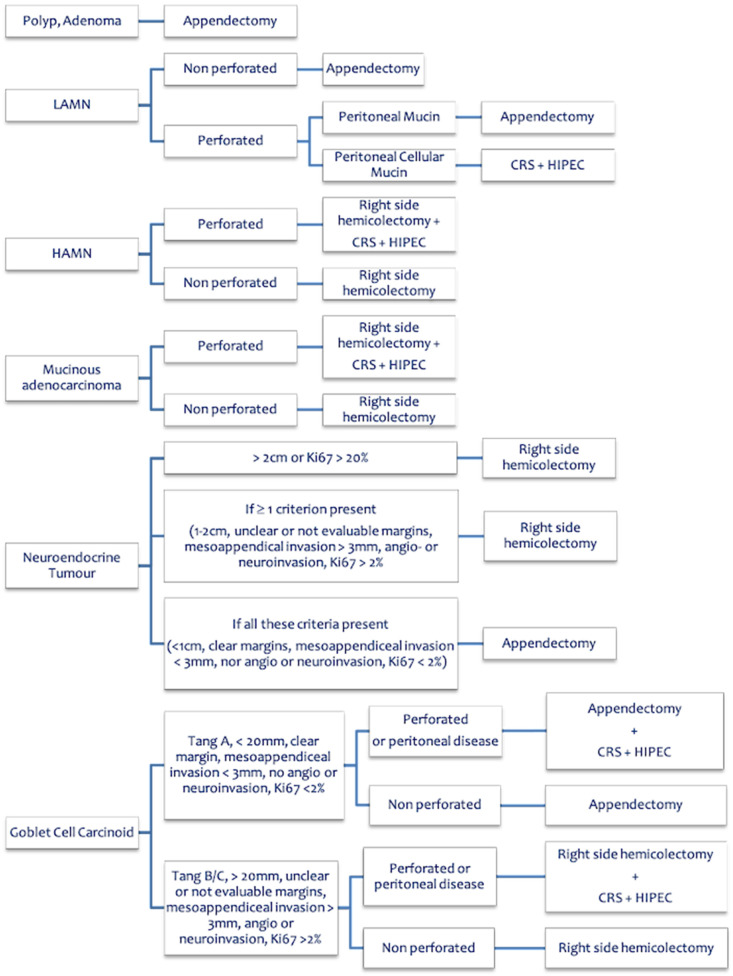
Primary appendiceal cancer treatment. LAMN = Low-grade Appendiceal Mucinous Neoplasm; HAMN = High grade Appendiceal Mucinous Neoplasm; CRS = Cytoreductive Surgery; HIPEC = Hyperthermic IntraPEritoneal Chemotherapy.

**Table 1 cancers-15-00728-t001:** Statements for consensus.

Patients with suspected or diagnosed PMP should be evaluated by a peritoneal tumor referral center.
2. In case of an unexpected finding of PMP during emergency surgery, surgical procedures must be limited to treating the emergency and obtaining biopsies for a histopathological diagnosis.
3. If PMP is discovered unexpectedly during abdominal surgery, surgical procedures must be limited to biopsies or appendectomy to obtain a histopathological diagnosis.
4. PSOGI 2016 histopathological classification of PMP and appendiceal neoplasia should be adopted.
5. Preoperative determination of serum CEA (Carcinoembryonic Agent) and CA (Carbohydrate Antigen) 19.9 must be performed.
6. Preoperative determination of serum CA (Carbohydrate Antigen) 125 could be performed.
7. A CT scan with contrast enhancement represents the gold standard for staging patients with PMP.
8. In patients with appendiceal PMP, a preoperative colonoscopy (to exclude second primaries and exclude the invasion of the appendicular stump) should be performed.
9. Laparoscopic evaluation could be included in the preoperative work-up of patients with PMP in order to obtain a histopathological diagnosis and/or evaluate resectability.
10. In cases of a clear diagnosis of PMP based on radiological imaging, laboratory testing, and clinical presentation, histopathological diagnostic confirmation prior to therapeutic decision-making should be conducted.
11. A histological review of the specimens of a patient with an appendiceal neoplasm or PMP by a pathologist expert in peritoneal surface malignancies must always be performed.
12. In the case of an adenoma or hyperplastic polyp with a clear margin, the appendectomy is curative.
13. In cases where a non-perforated LAMN is discovered after appendectomy, follow-up is indicated.
14. In cases of stump involvement on histopathological examination after appendectomy for LAMN, a cecotomy/ileocecal resection should be considered.
15. In cases where a perforated LAMN or extra-appendiceal mucin are discovered after appendectomy, cytoreductive surgery, and HIPEC could be considered as treatment option.
16. In cases where a non-perforated HAMN is found after appendectomy, a right-sided hemicolectomy should be considered.
17. In cases where a perforated HAMN and/or peritoneal disease are found after appendectomy, a right-sided hemicolectomy associated with CRS and HIPEC should be considered.
18. In cases where a non-perforated mucinous adenocarcinoma is discovered after appendectomy, a right-sided hemicolectomy should always be performed.
19. In cases where a perforated mucinous adenocarcinoma and/or peritoneal disease are discovered after appendectomy, a right-sided hemicolectomy with CRS and HIPEC is always recommended.
20. In the case of a neuroendocrine appendiceal tumor, if a carcinoid > 2 cm or a G3 proliferation rate (Ki67 > 20%) is present, a right-sided hemicolectomy should be performed.
21. In the case of a neuroendocrine appendiceal tumor, if one or more of the following features are present [tumor of 1–2 cm, positive or unclear margins, mesoappendiceal invasion >3 mm, vascular or lymphatic vessel invasion, G2 proliferation rate (Ki67 3–20%)], a right-sided hemicolectomy could be performed.
22. In the case of a goblet cell tumor classified as a Tang A lesion <20 mm with clear margins, mesoappendiceal invasion <3 mm with no vascular or lymphatic vessel involvement, and Ki67 <2%, appendectomy is curative.
23. In the case of a goblet cell tumor presenting one or more of the following features (Tang B-C lesion, >20 mm, unclear or non-evaluable margins, mesoappendiceal invasion >3 mm, vascular or lymphatic vessel involvement, Ki67 >2%) a right-sided hemicolectomy should be performed.
24. In cases of perforated goblet cell carcinoma or evidence of peritoneal spread, CRS + HIPEC could be considered.
25. In patients with PMP, CRS + HIPEC should be considered the first therapeutic option.
26. In patients with PMP, extensive small bowel involvement or mesenteric involvement inducing retraction should be considered absolute contraindications for CRS + HIPEC.
27. In patients with PMP submitted to CRS + HIPEC, no residual disease, or <2.5 mm should be obtained.
28. In patients with low-grade PMP submitted to complete CRS + HIPEC, adjuvant systemic chemotherapy should not be considered.
29. In patients with high-grade PMP/signet ring cells who have undergone complete CRS + HIPEC, adjuvant chemotherapy could be considered.
30. In patients with unresectable PMP who are not candidates for CRS + HIPEC, maximal tumor debulking in a specialized center may be considered.
31. In patients with PMP who present with unresectable disease or are not fit for CRS + HIPEC, systemic chemotherapy could be considered.
32. In patients with PMP presenting high-risk general conditions and borderline resectability, a “delayed” or “two-stage” CRS + HIPEC could be considered.
33. According to the literature, mitomycin C is recommended in patients with PMP who are candidates for CRS + HIPEC.
34. Every patient with a diagnosis of PMP or appendiceal mucinous neoplasm should be discussed in a dedicated multidisciplinary meeting.

**Table 2 cancers-15-00728-t002:** Results of consensus.

Statement	AMS	Appropriate	IPR	Results
1	9	Yes	9–9	valid
2	9	Yes	9–9	valid
3	9	Yes	9–9	valid
4	9	Yes	9–9	valid
5	9	Yes	9–9	valid
6	9	Yes	8–9	valid
7	9	Yes	9–9	valid
8	9	Yes	9–9	valid
9	9	Yes	8.7–9	valid
10	9	Yes	9–9	valid
11	9	Yes	8.7–9	valid
12	9	Yes	9–9	valid
13	9	Yes	8.7–9	valid
14	9	Yes	9–9	valid
15	9	Yes	9–9	valid
16	9	Yes	9–9	valid
17	9	Yes	9–9	valid
18	9	Yes	9–9	valid
19	9	Yes	9–9	valid
20	9	Yes	9–9	valid
21	9	Yes	8.7–9	valid
22	9	Yes	8.7–9	valid
23	9	Yes	9–9	valid
24	9	Yes	9–9	valid
25	9	Yes	9–9	valid
26	8	Yes	6.7–8	not valid
27	9	Yes	8.7–9	valid
28	9	Yes	9–9	valid
29	9	Yes	9–9	valid
30	9	Yes	8.7–9	valid
31	8	Yes	8–9	valid
33	9	Yes	8–9	valid
33	9	Yes	8.7–9	valid
34	9	Yes	9–9	valid
Flowchart 1	8	Yes	8–9	valid
Flowchart 2	9	Yes	8–9	valid

## Data Availability

Data and code for analysis are available upon request to the authors.

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
