# Peer review of "Diagnostic and Therapeutic Algorithm for Appendiceal Tumors and Pseudomyxoma Peritonei: A Consensus of the Peritoneal Malignancies Oncoteam of the Italian Society of Surgical Oncology (SICO)"

_cancers, 2023, doi:10.3390/cancers15030728_

Round 1

Reviewer 1 Report

The italian SICO group is well recommended internationally and present their recommendations on how to treat PMP and appendices malignancies in general. National (and international) guidelines are essential for uniform treatment and referral, and should, therefore, in general be published. Especially in a low evidence area as PMP, CRS and HIPEC. In general I applause the work made by the group. I cannot, and should not, change guidelines as they are italian recommendations made by italian experts. I have, however, some comments to the manuscript:

Line 56 "Jelly belly" is a term not appropriate in a scientific report. Personal opinion.

Line 66-67: "rarely causes...metasteses". This refers to PMP. However, just before, in line 61-62, the authors are writing about appendices cancers. 

77. A few more reasons have already been mentioned in the abstract: Standardize treatment and correct use of resources

87. RAND/UCLA abbreviation is not explained

203-204. This sentence does not concur with statement 33

Furthermore, the scientific basis for the statements should be provided as references, preferably for each statement either in the existing table with statements or in a separate table

Discussion: Differentiation between extraappendicular mucin vs. extraappendicular cells, and the importance hereof, shall be explained further. Will the difference influence the treatment algorithm?

Dicuussion. NET algorithm should be commented in relation to the ENETS consensus guideline

Author Response

Many thanks for suggestions, I made most of the corrections suggested. 

The statements came from a consensus, references are not possible for each statement, the base we moved on are cited in bibliogaphy.

The statement about mucin and cells in the discussion has been clarified

Reviewer 2 Report

The authors aimed at generationg recommendationf for the diagnosis and management of PMP in Italy based on expert consensus.  As the authors clearly stated in the manuscript, the low incidence and prevalence of this rare condition coupled with the clear advantage of CRS and HIPEC make it almost impossible / non-ethical to do a prospective randomized controllen trial to generate evidence. Thus a diagnostic and treatment algorithm based on consens from experts is a good way to tackle this difficulty.

The manuscript basically addresses all relevant issues on PMP and I my opinion could be a good tool for non-expert physicians.  Personally, i would have opted to included some images of these rares pathologies in the manuscript. The authors may want to consider this option.

Perforated appendix with LAMN may not always require CRS and HIPEC. I do think Appendektomie with partial peritonectomy in the right lower quadrant may a good option in selected cases. Could the authors please comment on this?

The manuscript would benefit from minor language polishing. 

Author Response

Many thanks for the suggestions. We made little change on the statement in the discussion about cells and mucin in order to clarify that CRS+HIPEC, as you suggested is not always the only  treatment options.